# Gene expression signatures between *Limia perugiae* (Poeciliidae) populations from freshwater and hypersaline habitats, with comparisons to other teleosts

**Elizabeth J. Wilson**[1]\*, **Nick Barts**[2], **John L. Coffin**[1], **James B. Johnson**[3], **Carlos M. Rodríguez Peña**[4], **Joanna L. Kelley**[5], **Michael Tobler**[6,7,8], **Ryan Greenway**[1]

1 Division of Biology, Kansas State University, Manhattan, KS, United States of America, 2 Department of Biology, University of Central Missouri, Warrensburg, MO, United States of America, 3 Divison of Marine Fisheries, North Carolina Department of Environmental Quality, Morehead City, NC, United States of America, 4 Instituto de Investigaciones Botánicas y Zoológicas, Universidad Autónoma de Santo Domingo, Santo Domingo, Dominican Republic, 5 Department of Ecology and Evolutionary Biology, University of California Santa Cruz, Santa Cruz, CA, United States of America, 6 Department of Biology, University of Missouri—St. Louis, St. Louis, MO, United States of America, 7 Whitney R. Harris World Ecology Center, University of Missouri—St. Louis, St. Louis, MO, United States of America, 8 WildCare Institute, Saint Louis Zoo, St. Louis, MO, United States of America

\* ejwilson2@ksu.edu

**Data Availability Statement:** All data is available on NCBI with the accession numbers listed in Table 1 of the manuscript. Accessions can be input

## Abstract

Salinity gradients act as strong environmental barriers that limit the distribution of aquatic organisms. Changes in gene expression associated with transitions between freshwater and saltwater environments can provide insights into organismal responses to variation in salinity. We used RNA-sequencing (RNA-seq) to investigate genome-wide variation in gene expression between a hypersaline population and a freshwater population of the livebearing fish species *Limia perugiae* (Poeciliidae). Our analyses of gill gene expression revealed potential molecular mechanisms underlying salinity tolerance in this species, including the enrichment of genes involved in ion transport, maintenance of chemical homeostasis, and cell signaling in the hypersaline population. We also found differences in gene expression patterns associated with cell-cycle and protein-folding processes between the hypersaline and freshwater *L. perugiae*. Bidirectional freshwater-saltwater transitions have occurred repeatedly during the diversification of fishes, allowing for broad-scale examination of repeatable patterns in evolution. Therefore, we compared transcriptomic variation in *L. perugiae* with other teleosts that have made freshwater-saltwater transitions to test for convergence in gene expression. Among the four distantly related population pairs from high- and low-salinity environments that we included in our analysis, we found only ten shared differentially expressed genes, indicating little evidence for convergence. However, we found that differentially expressed genes shared among three or more lineages were functionally enriched for ion transport and immune functioning. Overall, our results—in conjunction with other recent studies—suggest that different genes are involved in salinity transitions across disparate lineages of teleost fishes.

and retrieved from the NCBI database at the following link: https://www.ncbi.nlm.nih.gov/. On this page, the database should be set to "SRA" before searching for the accession numbers.

**Funding:** Funding was provided by National Science Foundation (IOS-1931657, IOS-2311366), the Army Research Office (W911NF-15-1-0175, W911NF-16-1-0225), and the Des Lee Collaborative Vision in Zoological Studies. The funders had no role in study design, data collection and analysis, decision to publish, or preparation of the manuscript.

**Competing interests:** The authors have declared that no competing interests exist.

## Introduction

Salinity variation imposes osmoregulatory challenges on aquatic organisms, and contact zones between freshwater and saltwater environments act as barriers that limit the ability of animals to move from one habitat to the other [1]. Many aquatic taxa have consequently failed to cross natural salinity gradients [2]. Those that achieve such habitat shifts overcome osmoregulatory challenges through plasticity or adaptation, and both responses have greatly shaped aquatic species distributions [3–5]. Due to the ecological expansion that accompanies colonization of novel habitats, invasions from freshwater to saltwater environments, or vice versa, that result in adaptation are of particular interest for elucidating the genetic mechanisms underlying salinity tolerance and the diversification of aquatic organisms [1, 2, 6].

### Salinity transitions in fishes

Among fishes, diversification has coincided with repeated transitions between freshwater and saltwater habitats [2]. Many saltwater-freshwater transitions in fishes have occurred over long evolutionary timescales, and deep evolutionary divergences have resulted in many species only tolerating a narrow range of salinities, restricting them to either freshwater or saltwater environments [2, 6–9]. A considerably smaller portion of fish species can survive in both freshwater and saltwater environments [10]. Among species that tolerate a broad range of salinities, movement along salinity clines is characteristic of diadromous lineages with life histories involving migration between freshwater and saltwater environments during an individual's lifetime [10, 11]. While some species cannot cross the saltwater-freshwater boundary and some do so routinely in their lifetimes, there are also lineages between these two extremes that have made transitions between saltwater and freshwater environments at microevolutionary scales [6, 12, 13]. Few studies have investigated mechanisms of salinity adaptation in species where closely related populations experience different salinity regimes, and the evolutionary repeatability of these mechanisms across lineages remains to be explored.

Transitions between freshwater and saltwater environments are challenging for animals that actively maintain internal solute homeostasis. The many physiological processes involved in stable state osmotic and ionic balance necessitate changes in multiple interdependent processes when crossing a salinity barrier [14]. To maintain homeostasis, fishes in freshwater must actively absorb salt and excrete water in the form of dilute urine to counteract their passive loss of salt and absorption of water [14]. In contrast, fishes in saltwater environments must remove salt and retain water [14]. As a result, crossing a salinity barrier requires a shift between absorption and excretion of ions and water in multiple organs, involving both active and passive processes [14, 15]. Remodeling of gill epithelia and regulation of ion transporters, aquaporins, and tight junctions in the gill are particularly central to this process [15–18].

### Gene expression responses to salinity variation

One way to quantify such complex physiological responses to variation in salinity is to compare patterns of gene expression across populations in different habitat types. Several studies have investigated the physiological and transcriptomic responses to changes in salinity between populations of the same species or among closely related species of fish [13, 18–20]. Differential gene expression associated with osmoregulation and ion transport is commonly found between populations inhabiting environments of different salinities [13, 18, 20]. In addition, differential expression has also been documented in genes associated with other biological functions, including immune processes [19, 20], cell communication [13], stress tolerance [13], and gill membrane permeability [19]. These studies highlight candidate pathways that play important roles in divergence between saltwater and freshwater ecotypes within species,

but there have been few comparisons investigating the repeatability of gene expression responses to variation in salinity across phylogenetically disparate taxa. Investigating evidence of convergence in gene expression patterns across taxa that have undergone similar salinity transitions will provide insight into possible shared and unique pathways involved in adaptation to different salinity regimes.

### Salinity tolerance in *Limia perugiae* (Poeciliidae)

In this study, we investigated patterns of gene expression in a freshwater and a hypersaline population of a livebearing fish species, *Limia perugiae* (Poeciliidae), and compared transcriptomic variation between these populations to those observed in other species that have undergone similar salinity transitions. Freshwater fishes in the genus *Limia* are endemic to the islands of the West Indies [21, 22]. *Limia perugiae* is a widespread species across the southern portion of Hispaniola, occurring in freshwater artesian springs and low-order creeks, as well as hypersaline inland lakes and coastal lagoons [23–25]. Exposure to high salinities in *L. perugiae* has been shown to decrease metabolic rate [25], increase the production of $Na^+/K^+$-ATPase and oxidative phosphorylation proteins in the gills [23], and reduce adult body size [23]. Though predominantly associated with freshwater habitats, many fishes in the family Poeciliidae are able to tolerate a broad range of salinities, a factor potentially responsible for facilitating their dispersal across a wide geographic range [26–29]. While the mechanisms and consequences of salinity tolerance at the biochemical and physiological levels have been a focus of research in poeciliids [23, 30–33], the genetic and regulatory underpinnings of salinity tolerance have yet to be investigated in this family. We used a natural system with conspecific populations occurring in both a freshwater and hypersaline habitat to characterize the potential molecular mechanisms underlying high salinity tolerance in poeciliid fishes.

### Objectives

We used RNA-sequencing (RNA-seq) to study genome-wide patterns of gene expression between freshwater and hypersaline *L. perugiae*. From this analysis, we aimed to identify genes and physiological pathways associated with salinity tolerance in this species. We then compared the *L. perugiae* population pair to other population pairs of freshwater and saltwater ecotypes in disparate actinopterygian taxa to understand if mechanisms of osmoregulatory capability are shared across divergent lineages of teleost fishes. We utilized a comparative transcriptomics approach that leverages new and pre-existing gene expression datasets to address the following questions: 1) What genes are differentially expressed between freshwater and saltwater *L. perugiae* populations, and with what physiological processes are they associated? 2) Is there evidence for commonalities in gene expression among phylogenetically disparate teleosts with freshwater and saltwater populations?

## Materials and methods

### Sample collection

*Limia perugiae* were collected using a seine from a hypersaline lagoon (Laguna Oviedo: 17.801˚N, 71.363˚W) and a geographically proximate freshwater stream (Los Cocos: 17.905˚N, 71.286˚W) in the Dominican Republic. Laguna Oviedo had a salinity of 54–61 ppt, and Los Cocos had a salinity of 0–1 ppt. Following capture, adult females ($N$ = 6 per site) were euthanized using MS222, and all efforts were taken to alleviate suffering during handling and euthanasia. Both sets of gills were extracted using sterilized scissors and forceps. Tissues were immediately preserved in RNAlater (Ambion Inc., Austin, TX, USA). All samples were

collected with permits from the corresponding authorities in the Dominican Republic (permit number 0092–11). All procedures involving animals followed established best-practices and were approved by the Institutional Animal Care and Use Committee of Kansas State University (protocol 3473).

### RNA-seq library preparation

RNA extraction, library preparation, and sequencing of samples followed procedures previously employed for related poeciliid species [34, 35]. Approximately 10–30 mg of tissue from each individual was sealed in a Covaris TT1 TissueTube (Covaris, Inc., Woburn, MA, USA), frozen in liquid nitrogen, and pulverized. Total RNA was extracted from the pulverized tissue using the NucleoSpin RNA kit (Machery-Nagel, Düren, Germany). mRNA isolation and cDNA library preparation were completed with the NEBNext Poly(A) mRNA Magnetic Isolation Module (New England Biolabs, Inc., Ipswich, MA, USA) and NEBNext Ultra Directional RNA Library Prep Kit for Illumina (New England Biolabs, Inc., Ipswich, MA, USA), with minor modifications to the manufacturers' protocol [34–36]. cDNA libraries were individually barcoded, quantified with Qubit and an Agilent 2100 Bioanalyzer High Sensitivity DNA chip, and then pooled with cDNA samples from other projects in sets of 11–12 samples based on nanomolar concentrations. Samples were split across pools such that samples from each habitat type were not all sequenced together, and there was no evidence for lane effects. Libraries were sequenced on an Illumina HiSeq 2500 using paired-end 101-base-pair (bp) reads at the Washington State University Spokane Genomics Core.

### Mapping

All raw reads were trimmed twice (quality 0 to remove Illumina adapters, followed by quality 24) using Trimgalore! v.0.4.0 [37]. Trimmed reads were mapped to the *Poecilia mexicana* reference genome (RefSeq accession number: GCF_001443325.1 [38]) with an appended mitochondrial genome (GenBank Accession Number: KC992998.1) using BWA-MEM v.0.7.12 [39]. We annotated genes from the *P. mexicana* reference genome by extracting the longest transcript for each gene (with the perl script gff2fast.pl: https://github.com/ISUgenomics/common_scripts/blob/master/gff2fasta.pl) and comparing them against entries in the human SWISS-PROT database (critical E-value 0.001; access date 04/15/2017) using BLASTx [40]. Each gene was annotated with the best BLAST hit from the human database based on the top high-scoring segment pair.

### Differential gene expression

We used StringTie (v.1.3.3b) [41, 42] to quantify the number of reads mapped to each gene for each individual (measured in counts per million mapped reads) and then used the prepDE.py script (provided with StringTie) to generate a read count matrix [42]. We removed genes that did not have at least two counts per million in three or more individuals across both populations, resulting in 18,659 genes that were included in differential gene expression analysis. We identified differentially expressed genes using generalized linear models (GLMs) in R, as implemented in the Bioconductor package edgeR [43]. We fit a negative binomial GLM to the normalized read counts of each gene based on tagwise dispersion estimates and a design matrix describing the comparison between the saltwater and freshwater population using glmFit. The tagwise dispersion estimates were generated using the estimateDisp function in edgeR, which employs a weighted likelihood empirical Bayes approach [44]. We assessed statistical significance using the GLM likelihood-ratio test with a false discovery rate (FDR) of q-value < 0.05, calculated with the Benjamini-Hochberg correction [45]. After identifying the

set of differentially expressed genes between the saltwater and freshwater population, we used a Gene Ontology (GO) enrichment analysis to explore putative biological functions of these genes. We annotated all differentially expressed genes that had a match in the human SWIS-S-PROT database with GO IDs [46] and tested for the enrichment of specific GO IDs separately in up and downregulated genes relative to the full background set of 18,659 genes using GOrilla (FDR q-value < 0.05, accessed May 26, 2022) [47]. A total of 10,935 genes in the background set were associated with a GO term in the database.

## Weighted gene co-expression network analysis

We constructed weighted gene co-expression networks to identify clusters of genes that were co-expressed across our samples [48]. To prepare the gene expression data for this analysis, we applied a variance-stabilizing transformation to the filtered reads using the varianceStabilizingTransformation function from the DESeq2 package (v.1.36.0) [49] in R, which allowed us to normalize the read counts relative to library size and appropriately scale the data for clustering [50, 51]. After transforming and normalizing the read counts, we used the cpm function in the edgeR package (v.3.38.1) [43] to generate a gene matrix of scaled read counts ($log_2$-cpm, counts per million mapped reads) from the transformed read counts [52].

As outlined in the Weighted Gene Co-expression Network Analysis (WGCNA) package documentation [50], we used hierarchical clustering to cluster the samples based on their gene expression profiles. We used the hclust function from the flashClust package (v.1.1.2) [51] to cluster the samples. We then created a weighted network adjacency matrix using the adjacency function from the WGCNA package (v.1.71) [50, 51]. The adjacency matrix was constructed by calculating pairwise co-expression similarities (Pearson correlation coefficients) and raising them to a power of β, a soft thresholding power [50]. We used the pickSoftThreshold function in the WGCNA package [50] to assist in selecting the lowest possible value of β that ensured our network fit the approximate scale free topology criterion while retaining the highest possible mean connectivity between the network genes. Based on the scale free topology model fit and mean connectivity of our network, and following the recommendations outlined by Zhang and Horvath [48], we selected β = 7.

From our correlation network, we then generated a topological overlap dissimilarity matrix to identify modules of co-expressed genes. We calculated dissimilarity between the genes by converting the adjacencies into topological overlap similarities using the TOMsimilarity function in the WGCNA package [50] and then subtracting these topological overlap measures from 1. To identify modules of co-expressed genes, we used the hclust function from the flashClust package [51] for hierarchical clustering of the genes and then created a hierarchical clustering dendrogram. We used the cutreeDynamic function from the dynamicTreeCut package [50, 53] to extract the modules from the dendrogram. To summarize the gene expression variation in each module, the first principal component of each module in the expression matrix (the eigengene) was calculated using the moduleEigengenes function from the WGCNA package [50]. We used the function mergeCloseModules in the WGCNA package to merge eigengenes that were highly correlated. We included eigengenes with a correlation greater than 0.9 for merging. Finally, we identified modules that were significantly associated with the presence or absence of salinity by calculating correlation coefficients between the eigengenes and the habitat type. *P*-values of the correlation coefficients were calculated using the corPvalueStudent function from the WGCNA package, and we retained modules with *P*-values less than 0.01 for functional enrichment analyses. Similar to our differential gene expression analysis, we used GOrilla [47] for Gene Ontology enrichment analysis of genes contained in modules exhibiting significant correlations with habitat type.

## Comparisons of *L. perugiae* with other species

We mined previously published datasets to identify gene expression patterns commonly associated with salinity tolerance in disparate taxa, including South American silversides (*Odontesthes bonariensis* and *Odontesthes argentinensis*) from Hughes et al. [20], three-spine sticklebacks (*Gasterosteus aculeatus*) from Gibbons et al. [19], Amur ide (*Leuciscus waleckii*) from Xu et al. [13], and *L. perugiae* (this study; Table 1). Each of these experiments generated paired-end RNA-seq raw reads from gill tissue in two ecotypes (one freshwater and the other saltwater) of the same lineage.

Raw RNA-seq reads from each transcriptomics project were downloaded in FASTQ format, and reference genomes or transcriptomes for each species were downloaded in FASTA format from Genbank (see Table 1 for accession numbers). Reads were trimmed and mapped to their respective reference genomes (*Poecilia mexicana*: GCF_001443325.1 [38]; *Cyprinus carpio*: GCF_000951615.1 [54]; *Gasterosteus aculeatus*: Broad S1 v. 93 [55]) or reference transcriptome (*Menidia menidia*: GEVY00000000.1 [56]) following the same methods described above for *L. perugiae*. We then quantified the number of reads mapped to each gene in the annotation file for each reference genome and created a read counts matrix for each species, which were used for subsequent expression analyses. Expression analyses were performed in R version 4.1.2. The 10,000 genes with the highest standard deviation between freshwater and saltwater samples were abstracted from each read counts matrix, and overall expression patterns were visualized with multi-dimensional scaling (MDS) plots.

To make comparisons across species, we used OrthoFinder v2.2.6 to identify orthologous genes among the reference genomes [57, 58]. For OrthoFinder, we used the 'dendroblast' option for gene tree inference, 'blast' for the sequence search program, 'mafft' for the multiple sequence alignment, and 'fasttree' for the tree inference method. A total of 18,419 orthogroups were identified. Of the total orthogroups identified, 13,899 orthogroups had at least one copy in each species. Of those, 1,735 were 1:1 orthologs. To calculate counts per orthogroup, we used the gene counts matrix of each species to sum up the counts across all loci contained in an orthogroup. Based on this orthogroup counts matrix, we retained only the orthogroups that were present in all species and expressed in all individuals (cpm > 0 per individual), resulting in 12,743 retained orthogroups.

**Table 1. Species included in the analysis, including environment (freshwater [FW] or saltwater [SW]), collection location, sample size (N), NCBI Sequence Read Archive (SRA) accession numbers, and study reference.**

| Species | Environment | Collection Location | N | SRA Accessions | Study Reference |
|---|---|---|---|---|---|
| *Limia perugiae* | FW | Los Cocos (17.905˚N, 71.286˚W), Dominican Republic | 6 | SRX20992238, SRX20992239, SRX20992242, SRX20992243, SRX20992244, SRX20992245 | This study |
| *Limia perugiae* | SW | Laguna Oviedo (17.801˚N, 71.363˚W), Dominican Republic | 6 | SRX20992246, SRX24462927, SRX20992247, SRX20992248, SRX20992240, SRX20992241 | This study |
| *Gasterosteus aculeatus* | FW | Trout Lake (49˚30029″N, 123˚52029″W), British Columbia, Canada | 5 | SRX2544970, SRX2544969, SRX2544968, SRX2544967, SRX2544966 | [19] |
| *Gasterosteus aculeatus* | SW | Oyster Lagoon (49˚36043.53″N, 124˚01052.12″W), British Columbia, Canada | 5 | SRX2544985, SRX2544984, SRX2544983, SRX2544982, SRX2544981 | [19] |
| *Odontesthes bonariensis* | FW | Lake Chascomús (35˚34′S, 58˚01′W), Argentina | 3 | SRX1681471, SRX1681473, SRX1681474 | [20] |
| *Odontesthes argentinensis* | SW | Mar del Plata (38˚02′S, 57˚31′W), Argentina | 3 | SRX1671790, SRX1681012, SRX1681017 | [20] |
| *Leuciscus waleckii* | FW | Ganggeng Nor Lake (43˚17'48″N, 116˚53'27″E), Mongolia | 1 | SRX333071 | [13] |
| *Leuciscus waleckii* | SW | Dali Nor Lake (43˚22′43″N, 116˚39'24″E), Mongolia | 1 | SRX1410650 | [13] |

To evaluate expression differences for each orthogroup, we made pairwise comparisons between ecotypes following the same methods described above for the *L. perugiae* comparisons. Briefly, we normalized reads, created and compared generalized linear models of the normalized read counts, generated a design matrix, estimated tagwise dispersion, and conducted GLM likelihood-ratio tests to test whether differences in expression were statistically different between the freshwater and saltwater population for each orthgroup. To identify orthogroups exhibiting convergent expression patterns across lineages, we intersected the significantly upregulated and downregulated orthogroups from all lineage-specific comparisons, identifying orthogroups that were differentially expressed in the same direction in pairwise, three-, and four-way comparisons among the lineages. After identifying the set of orthogroups with differential expression across three or more lineages, we used a GO enrichment analysis as described above to explore the putative biological functions of these candidate gene sets.

## Results

### Comparative analysis of freshwater and hypersaline *L. perugiae*

We used RNA-seq to characterize the transcriptomes of *L. perugiae* from a freshwater (n = 6) and a hypersaline populations (Table 1). 73,590,325 total raw reads were obtained across all individuals: 34,998,157 from freshwater *L. perugiae* (n = 6) and 38,592,168 from saltwater *L. perugiae* (n = 6) before trimming (Table A in S1 Appendix). After trimming, 95.5% of reads from the freshwater individuals mapped to the *Poecilia mexicana* reference genome, and 95.2% mapped for the saltwater individuals (Table A in S1 Appendix).

We identified 4,895 differentially expressed genes between saltwater and freshwater ecotypes of *L. perugiae*, 2,437 of which were upregulated and 2,458 of which were downregulated in the saltwater ecotype (Fig 1A). The genes upregulated in the saltwater population were largely associated with ion transport, maintaining chemical homeostasis, and cell signaling (FDR < 0.05) (Table 2 and S1 Fig). Processes relevant to chemical homeostasis included several solute carrier genes, such as *SLC9A3*, *SLC8B1*, *SLC12A8*, and *SLC30A9*. $Na^+/K^+$-ATPase and other ATPase genes, such as *ATP1B1* and *ATP6V1A*, were also upregulated among the genes involved in chemical homeostasis and signal transduction. Genes downregulated in the saltwater population corresponded to GO process terms such as mitotic cell cycle process, protein folding, chromosome segregation, rRNA processing, and mitochondrial translational elongation (FDR < 0.05; Table 2 and S2 Fig).

Weighted gene co-expression network analysis (WGCNA) revealed five modules of co-expressed genes that were significantly correlated with salinity (Fig 2). The turquoise module was positively correlated with salinity (*P*-value < 0.01), and the black, red, royalblue, and blue modules were all negatively correlated with salinity (Fig 2). Out of the 18,659 genes included in our analysis, the positively correlated module (turquoise module) contained 4,056 genes. The negatively correlated modules contained 2,166 genes (black module), 837 genes (red module), 173 genes (royalblue module), and 3,300 genes (blue module). The correlation coefficients and their associated *P*-values between each gene and the environmental condition (salinity), and between each gene and each module, are included in Table B in S1 Appendix. Each gene's module assignment can also be found in Table B in S1 Appendix. From the functional enrichment analysis, we found that the turquoise, royalblue, and blue modules were significantly enriched for biological processes, and these modules of co-expressed genes largely corroborated the differential expression results. Like the biological processes that were enriched among the up-regulated genes in the saltwater population, the module that was positively correlated with salinity (turquoise) was functionally enriched for GO terms involved in ion transport and cell signaling (Table C in S1 Appendix). There were, however, GO terms

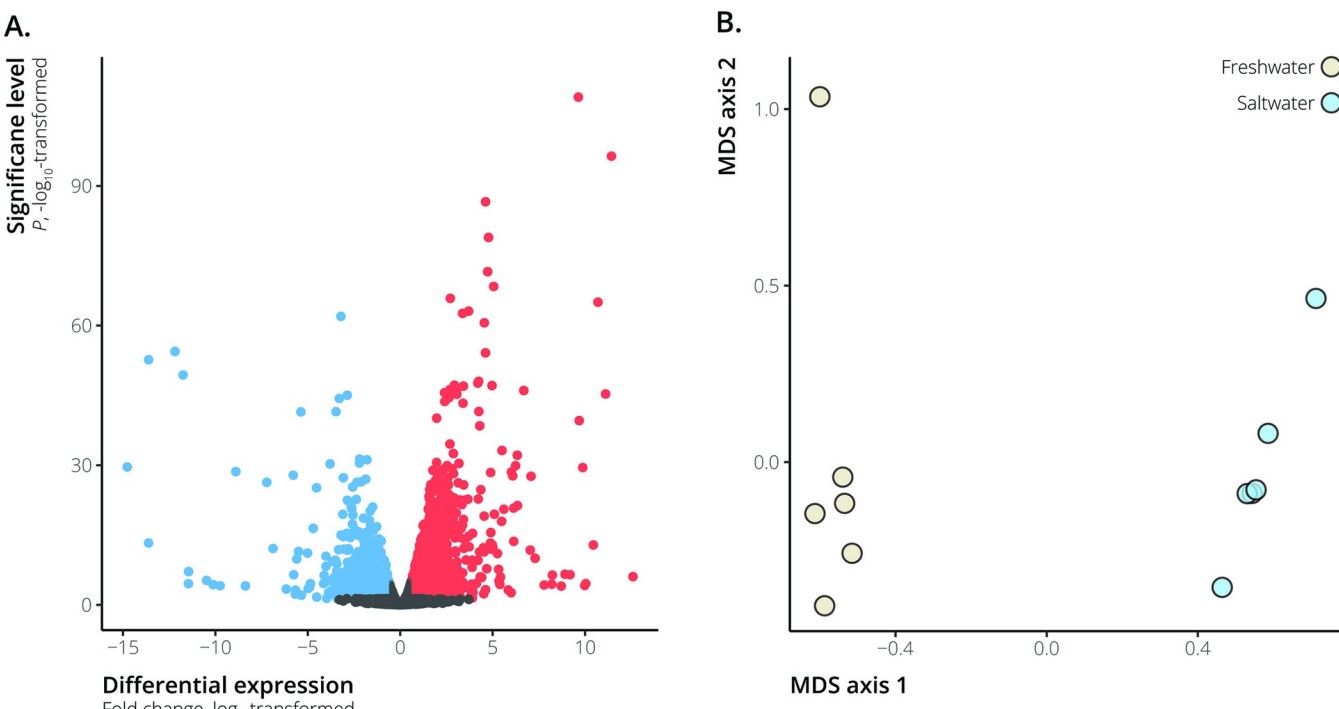

**Fig 1. Differentially expressed genes and gene expression profiles of hypersaline and freshwater *Limia perugiae*.** A. Volcano plot depicting differentially expressed genes between hypersaline and freshwater *Limia perugiae*. Genes that were significantly differentially expressed between hypersaline and freshwater populations (FDR < 0.05) are indicated by the blue and red points—blue points represent genes downregulated in the hypersaline populations, while red points represent upregulated genes. B. Multi-dimensional scaling (MDS) plot of hypersaline and freshwater *L. perugiae* gene expression profiles. MDS axis 1 separated samples by freshwater vs. hypersaline environments.

associated with the turquoise module that were not identified from the differential expression analysis, including regulation of autophagy and lipid transport (Table C in S1 Appendix). The modules that were negatively correlated with salinity also reflected biological processes that were associated with down-regulated genes in the saltwater *L. perugiae*, including regulation of the cell cycle and protein folding (Tables D and E in S1 Appendix). The royalblue module only had one significantly enriched GO term, chemokine-mediated signaling pathway, which was a GO term also associated with down-regulation of genes in the saltwater environment (Table D in S1 Appendix). In contrast, the blue module contained several significant GO terms, including mitotic cell cycle process, protein folding, rRNA metabolic process, and chromosome organization (Table E in S1 Appendix). There were many GO terms associated with the blue module that were not represented in the differential expression analysis, including sarcomere organization, macromolecule biosynthetic process, regulation of cellular response to heat, and nucleic acid metabolic process (Table E in S1 Appendix). Several of the GO terms unique to the blue module were related to cellular metabolism and biosynthesis pathways (Table E in S1 Appendix).

## Comparisons of *Limia* with phylogenetically disparate teleosts

We compared transcriptomic differences between *L. perugiae* populations to previously published transcriptome data from teleosts with populations from freshwater and saline habitats (Table 1). Mapping statistics for all four population pairs can be found in Table A in S1 Appendix. As expected, MDS plots indicated that orthogroup expression variation was primarily

**Table 2. GO process terms with significant enrichment in genes upregulated and downregulated in the hypersaline ecotype of *Limia perugiae* (FDR < 0.05).**

| GO term | Description | Enrichment | N | P-value | FDR q-value |
|---|---|---|---|---|---|
| **Upregulated** | | | | | |
| GO:0006820 | anion transport | 1.8 | 100 | 5.67E-10 | 8.04E-06 |
| GO:0006811 | ion transport | 1.49 | 181 | 3.41E-09 | 2.42E-05 |
| GO:0007165 | signal transduction | 1.22 | 540 | 4.67E-09 | 2.21E-05 |
| GO:0051049 | regulation of transport | 1.36 | 242 | 4.50E-08 | 1.59E-04 |
| GO:0034220 | ion transmembrane transport | 1.51 | 129 | 3.41E-07 | 9.67E-04 |
| GO:0043269 | regulation of ion transport | 1.61 | 98 | 3.89E-07 | 9.19E-04 |
| GO:0010959 | regulation of metal ion transport | 1.85 | 62 | 4.53E-07 | 9.18E-04 |
| GO:0098656 | anion transmembrane transport | 1.89 | 57 | 5.26E-07 | 9.33E-04 |
| GO:0032879 | regulation of localization | 1.24 | 360 | 1.19E-06 | 1.87E-03 |
| GO:0015711 | organic anion transport | 1.71 | 73 | 1.21E-06 | 1.71E-03 |
| GO:0002028 | regulation of sodium ion transport | 2.72 | 23 | 1.62E-06 | 2.09E-03 |
| GO:0023051 | regulation of signaling | 1.19 | 451 | 3.07E-06 | 3.62E-03 |
| GO:0048878 | chemical homeostasis | 1.4 | 153 | 3.61E-06 | 3.94E-03 |
| GO:0010646 | regulation of cell communication | 1.19 | 444 | 7.08E-06 | 7.17E-03 |
| GO:0003254 | regulation of membrane depolarization | 3.24 | 15 | 7.84E-06 | 7.41E-03 |
| GO:0051050 | positive regulation of transport | 1.41 | 132 | 1.38E-05 | 1.23E-02 |
| GO:1902305 | regulation of sodium ion transmembrane transport | 2.85 | 17 | 1.76E-05 | 1.46E-02 |
| GO:0050794 | regulation of cellular process | 1.07 | 1144 | 2.50E-05 | 1.97E-02 |
| GO:0031344 | regulation of cell projection organization | 1.43 | 111 | 3.54E-05 | 2.64E-02 |
| GO:0043270 | positive regulation of ion transport | 1.78 | 46 | 3.61E-05 | 2.56E-02 |
| GO:0007186 | G protein-coupled receptor signaling pathway | 1.47 | 97 | 3.72E-05 | 2.51E-02 |
| GO:0010975 | regulation of neuron projection development | 1.51 | 84 | 3.89E-05 | 2.51E-02 |
| GO:0045664 | regulation of neuron differentiation | 1.44 | 105 | 4.04E-05 | 2.49E-02 |
| GO:0042908 | xenobiotic transport | 3.3 | 12 | 5.15E-05 | 3.04E-02 |
| GO:0019932 | second-messenger-mediated signaling | 1.69 | 53 | 5.17E-05 | 2.93E-02 |
| GO:0042592 | homeostatic process | 1.29 | 197 | 5.18E-05 | 2.83E-02 |
| GO:0120035 | regulation of plasma membrane bounded cell projection organization | 1.42 | 109 | 5.79E-05 | 3.04E-02 |
| GO:0006814 | sodium ion transport | 2.04 | 29 | 7.33E-05 | 3.71E-02 |
| GO:0032501 | multicellular organismal process | 1.19 | 360 | 7.62E-05 | 3.72E-02 |
| GO:0045216 | cell-cell junction organization | 1.94 | 33 | 7.65E-05 | 3.62E-02 |
| GO:0050767 | regulation of neurogenesis | 1.38 | 122 | 7.78E-05 | 3.56E-02 |
| GO:0055085 | transmembrane transport | 1.3 | 165 | 9.84E-05 | 4.36E-02 |
| GO:0030001 | metal ion transport | 1.47 | 85 | 1.05E-04 | 4.52E-02 |
| GO:0031345 | negative regulation of cell projection organization | 1.83 | 37 | 1.07E-04 | 4.46E-02 |
| GO:0065008 | regulation of biological quality | 1.15 | 453 | 1.21E-04 | 4.91E-02 |
| GO:0015849 | organic acid transport | 1.69 | 47 | 1.25E-04 | 4.91E-02 |
| GO:0046942 | carboxylic acid transport | 1.69 | 47 | 1.25E-04 | 4.77E-02 |
| GO:0065007 | biological regulation | 1.06 | 1274 | 1.27E-04 | 4.75E-02 |
| **Downregulated** | | | | | |
| GO:1903047 | mitotic cell cycle process | 1.66 | 149 | 2.52E-11 | 3.58E-07 |
| GO:0006457 | protein folding | 2.14 | 64 | 3.03E-10 | 2.15E-06 |
| GO:0022402 | cell cycle process | 1.46 | 191 | 6.63E-09 | 3.13E-05 |
| GO:0051301 | cell division | 1.71 | 98 | 1.40E-08 | 4.98E-05 |
| GO:0006364 | rRNA processing | 1.91 | 65 | 4.51E-08 | 1.28E-04 |
| GO:0009987 | cellular process | 1.05 | 1687 | 6.13E-08 | 1.45E-04 |
| GO:0016072 | rRNA metabolic process | 1.82 | 69 | 1.35E-07 | 2.73E-04 |

*(Continued)*

**Table 2.** (Continued)

| GO term | Description | Enrichment | N | P-value | FDR q-value |
|---|---|---|---|---|---|
| GO:0030198 | extracellular matrix organization | 1.75 | 76 | 2.13E-07 | 3.77E-04 |
| GO:0007059 | chromosome segregation | 2.38 | 33 | 3.54E-07 | 5.57E-04 |
| GO:0043062 | extracellular structure organization | 1.7 | 80 | 3.85E-07 | 5.46E-04 |
| GO:0000070 | mitotic sister chromatid segregation | 3.61 | 15 | 7.56E-07 | 9.74E-04 |
| GO:0000819 | sister chromatid segregation | 3.41 | 16 | 1.01E-06 | 1.19E-03 |
| GO:0007051 | spindle organization | 2.04 | 42 | 1.40E-06 | 1.53E-03 |
| GO:0043933 | protein-containing complex subunit organization | 1.28 | 269 | 2.07E-06 | 2.10E-03 |
| GO:0098813 | nuclear chromosome segregation | 3.14 | 17 | 2.34E-06 | 2.21E-03 |
| GO:0006415 | translational termination | 2.16 | 35 | 2.50E-06 | 2.22E-03 |
| GO:0043624 | cellular protein complex disassembly | 1.99 | 42 | 3.07E-06 | 2.56E-03 |
| GO:0070125 | mitochondrial translational elongation | 2.18 | 33 | 3.71E-06 | 2.92E-03 |
| GO:0070098 | chemokine-mediated signaling pathway | 3.17 | 16 | 4.01E-06 | 2.99E-03 |
| GO:0018208 | peptidyl-proline modification | 2.65 | 21 | 6.13E-06 | 4.34E-03 |
| GO:0007010 | cytoskeleton organization | 1.38 | 153 | 8.67E-06 | 5.85E-03 |
| GO:0007088 | regulation of mitotic nuclear division | 1.89 | 42 | 1.29E-05 | 8.34E-03 |
| GO:0044772 | mitotic cell cycle phase transition | 1.66 | 62 | 1.72E-05 | 1.06E-02 |
| GO:0006336 | DNA replication-independent nucleosome assembly | 3.1 | 14 | 2.24E-05 | 1.32E-02 |
| GO:0034622 | cellular protein-containing complex assembly | 1.36 | 150 | 2.47E-05 | 1.40E-02 |
| GO:0051783 | regulation of nuclear division | 1.82 | 44 | 2.48E-05 | 1.35E-02 |
| GO:0051383 | kinetochore organization | 4.16 | 9 | 2.59E-05 | 1.36E-02 |
| GO:0044770 | cell cycle phase transition | 1.63 | 63 | 2.67E-05 | 1.35E-02 |
| GO:0000278 | mitotic cell cycle | 1.94 | 36 | 2.98E-05 | 1.46E-02 |
| GO:0000413 | protein peptidyl-prolyl isomerization | 2.77 | 16 | 3.82E-05 | 1.80E-02 |
| GO:0034724 | DNA replication-independent nucleosome organization | 2.98 | 14 | 4.05E-05 | 1.85E-02 |
| GO:0070126 | mitochondrial translational termination | 2.02 | 31 | 4.10E-05 | 1.82E-02 |
| GO:0051276 | chromosome organization | 1.53 | 75 | 5.71E-05 | 2.45E-02 |
| GO:0071840 | cellular component organization or biogenesis | 1.11 | 739 | 6.92E-05 | 2.88E-02 |
| GO:0006414 | translational elongation | 1.85 | 37 | 7.46E-05 | 3.02E-02 |
| GO:0022610 | biological adhesion | 1.35 | 134 | 8.31E-05 | 3.27E-02 |
| GO:0007155 | cell adhesion | 1.35 | 133 | 8.91E-05 | 3.41E-02 |
| GO:0042493 | response to drug | 1.37 | 119 | 9.53E-05 | 3.56E-02 |
| GO:0016074 | snoRNA metabolic process | 3.46 | 10 | 9.90E-05 | 3.60E-02 |
| GO:0000281 | mitotic cytokinesis | 2.43 | 18 | 1.11E-04 | 3.95E-02 |
| GO:0000086 | G2/M transition of mitotic cell cycle | 1.83 | 36 | 1.15E-04 | 3.97E-02 |
| GO:1902850 | microtubule cytoskeleton organization involved in mitosis | 1.89 | 33 | 1.15E-04 | 3.88E-02 |
| GO:0045943 | positive regulation of transcription by RNA polymerase I | 3.21 | 11 | 1.16E-04 | 3.84E-02 |
| GO:0006334 | nucleosome assembly | 2.24 | 21 | 1.33E-04 | 4.30E-02 |
| GO:0044839 | cell cycle G2/M phase transition | 1.81 | 36 | 1.42E-04 | 4.46E-02 |
| GO:0072321 | chaperone-mediated protein transport | 4.31 | 7 | 1.58E-04 | 4.86E-02 |

The table includes the GO term ID, description, the degree of enrichment, the number of differentially expressed genes associated with the GO term (*N*), as well as *P* and FDR-corrected *q*-values.

driven by differences among taxonomic groups, with much smaller differences between ecotypes within species (Fig 3A). We then compared orthogroup expression profiles of all the freshwater and saltwater lineages based on mean expression values and found that the variation in orthogroup expression largely reflects phylogenetic divergence among lineages (Fig 3B). Closely related lineages exhibited more similar expression profiles, irrespective of

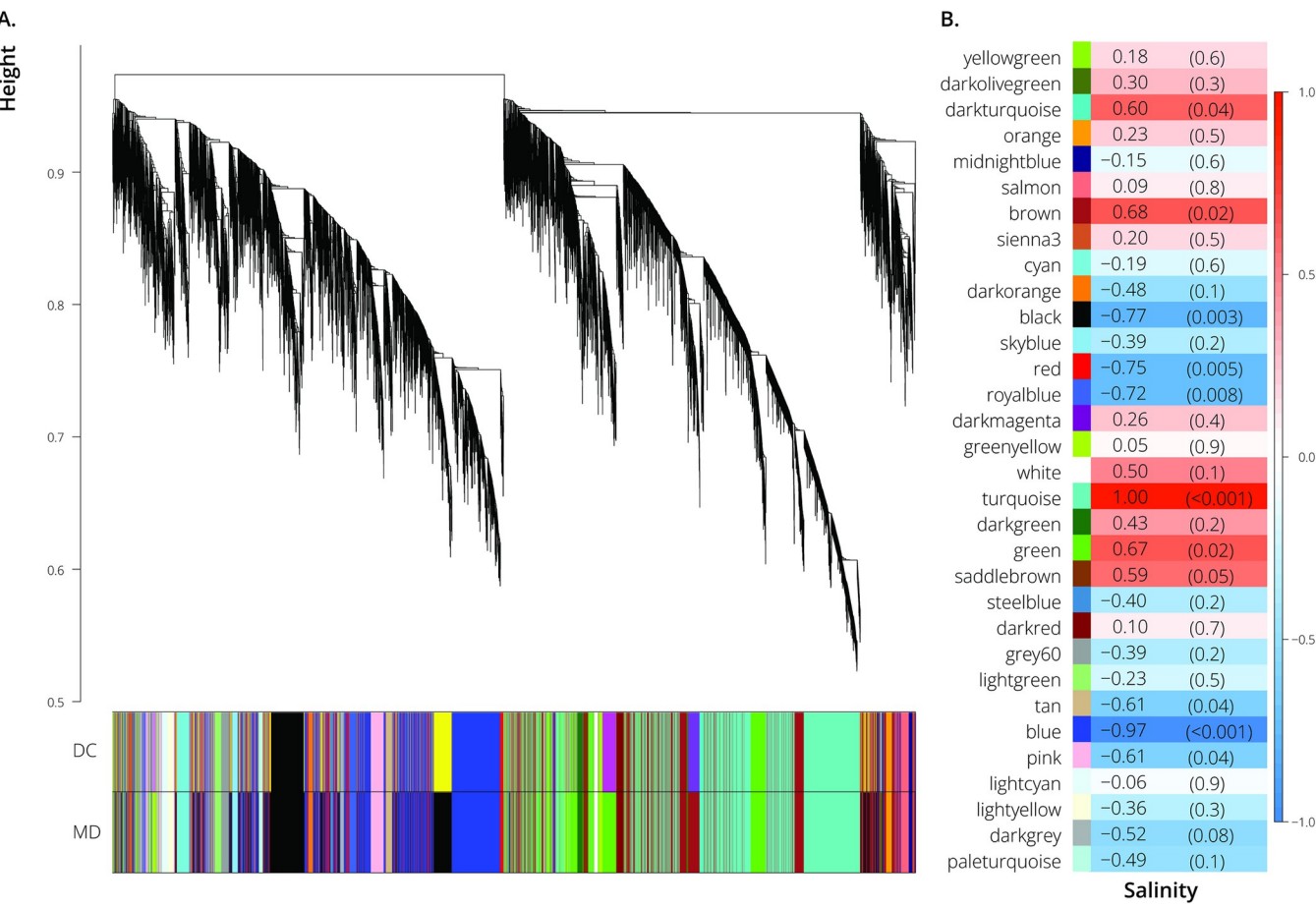

**Fig 2. Weighted gene co-expression network analysis.** A. Average linkage clustering tree based on topological overlap distances in gene expression patterns of *L. perugiae* from freshwater and saltwater habitats. Branches of the dendrogram correspond to modules, as shown in the color bars below. DC is an abbreviation for Dynamic Tree Cut; MD for Merged Dynamic. B. Correlation between module eigenvalues and habitat type (freshwater *vs.* saltwater). Each row corresponds to a module of coexpressed genes, and values are Pearson correlation coefficients (left column) and *P*-values (right column in parentheses). Color coloration scales with the correlation coefficient according to the scale bar to the right.

environmental conditions (saltwater vs. freshwater; Fig 3B). Across all groups, 122 differentially expressed orthogroups were shared across at least three lineages, but only 10 shared orthogroups were differentially expressed across all freshwater and saltwater population pairs (Fig 3C). Of those 10 orthogroups, 9 had annotations in the SWISS-PROT database (Table 3 and Fig 4), including a Na$^+$/H$^+$-exchanger (*SLC9A3*) involved in osmoregulation. The directionality of differential expression varied among lineages, and none of the 10 shared differentially expressed orthogroups were up-regulated or down-regulated among all four population pairs (Fig 4).

To investigate the biological processes reflected in shared differentially expressed orthogroups, we analyzed the differentially expressed orthogroups that were shared among three or more lineages (Fig 3C). GO analysis indicated enrichment in biological processes associated mostly with transmembrane transport (particularly ion transport) and some associated with immune function, which were all significant based on *P*-value (*P*-value < 0.001) but not after FDR correction (Table 4 and S3 Fig). Some of the GO processes specific to transmembrane transport included anion transport, inorganic cation import across plasma membrane, potassium ion import, urea transmembrane transport, and pyruvate transmembrane transport

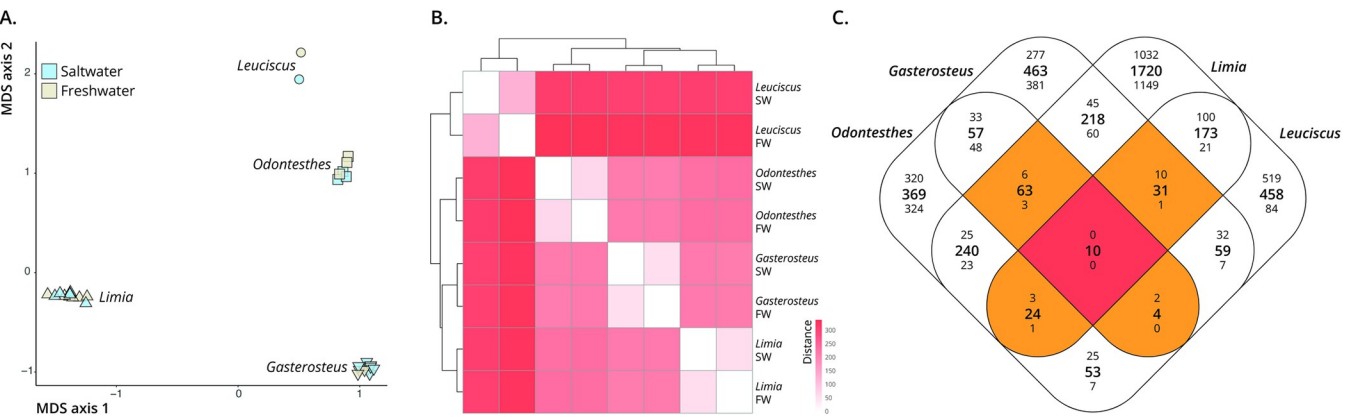

**Fig 3. Gene expression profiles and shared differentially expressed genes across lineages.** A. Multi-dimensional scaling plot (MDS) of the general expression patterns of all the populations included in our analysis. B. Similarity of gene expression profiles of saltwater (SW) and freshwater (FW) populations across different lineages. The majority of variation in gene expression reflects phylogenetic divergence among lineages. C. Shared differentially expressed genes across lineages. The large, central number in each section represents the total number of shared differentially expressed genes among the lineages in that intersection. The top number in each section represents the number of shared up-regulated genes, and the bottom number is the number of shared down-regulated genes in that intersection. Only 10 genes were consistently differentially expressed between all of the SW and FW populations.

($P$-value < 0.001; FDR > 0.05). The shared differentially expressed immune genes were reflective of negative regulation of T-helper cell differentiation, negative regulation of leukocyte differentiation, negative regulation of CD4-positive, alpha-beta T cell differentiation, and regulation of adaptive immune response based on somatic recombination of immune receptors built from immunoglobulin superfamily ($P$-value < 0.001; FDR > 0.05).

## Discussion

In this study, we used an RNA-seq approach to better understand mechanisms of salinity tolerance in the livebearing fish *Limia perugiae* (Poeciliidae), as well as to investigate whether mechanisms of osmoregulation are shared across distantly related species of teleosts that have all made recent transitions between freshwater and saltwater environments. We compared patterns of gene expression between a freshwater and a hypersaline population of *L. perugiae* and found that the genes upregulated in the saltwater population were largely related to ion transport and maintaining chemical homeostasis, while downregulated genes were associated with

**Table 3. Shared differentially expressed genes among all four population pairs and their associated proteins.**

| Protein name | Gene name |
| --- | --- |
| Sodium/hydrogen exchanger 3 | *SLC9A3* |
| Sulfide:quinone oxidoreductase, mitochondrial | *SQRDL* |
| Cytochrome b reductase 1 | *CYBRD1* |
| Zinc finger protein 800 | *ZNF800* |
| Pulmonary surfactant-associated protein D | *SFTPD* |
| Glucose-fructose oxidoreductase domain-containing protein 1 | *GFOD1* |
| Carcinoembryonic antigen-related cell adhesion molecule 1 | *CEACAM1* |
| 1-phosphatidylinositol 4,5-bisphosphate phosphodiesterase delta-1 | *PLCD1* |
| Cirhin | *CIRH1A* |

Based on the SWISS-PROT database, nine of the ten genes have experimental evidence for the existence of the protein associated with each gene. The tenth gene did not have a name match in the database, so it is not included.

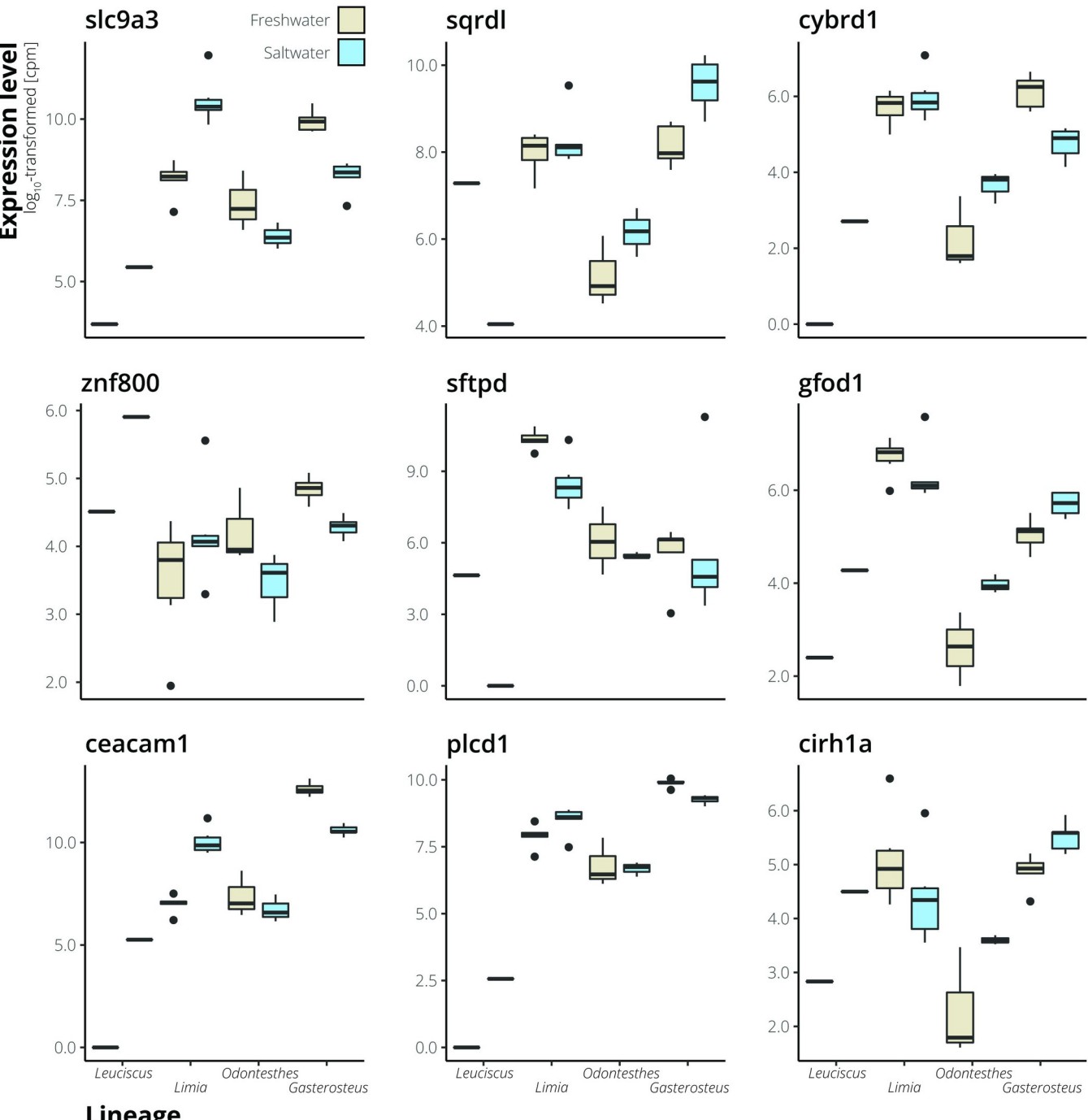

**Fig 4. Examples of expression variation in shared differentially expressed genes.** Nine of the ten shared differentially expressed genes have annotations, and those genes are included here. Magnitude and direction of differential expression is lineage specific.

processes involved in the cell cycle regulation and protein folding. These results provide insight into how *L. perugiae* has colonized a novel environment and maintains homeostasis under extreme salinity stress. Comparisons of our *Limia* results with pre-existing gene expression data collected from freshwater and saltwater ecotypes of South American silversides (*Odontesthes* spp.) [20], three-spine stickleback (*Gasterosteus aculeatus*) [19], and Amur ide

**Table 4. GO process terms that had significant enrichment in orthogenes with significant differential expression in at least three lineages (*P*-value < 0.001), but not after FDR correction (FDR > 0.05).**

| GO term | Description | Enrichment | N | *P*-value | FDR *q*-value |
|---|---|---|---|---|---|
| GO:0006820 | anion transport | 3.63 | 14 | 2.81E-05 | 3.71E-01 |
| GO:0098656 | anion transmembrane transport | 4.99 | 10 | 2.90E-05 | 1.91E-01 |
| GO:0099587 | inorganic ion import across plasma membrane | 10.07 | 5 | 1.17E-04 | 5.16E-01 |
| GO:0098659 | inorganic cation import across plasma membrane | 10.07 | 5 | 1.17E-04 | 3.87E-01 |
| GO:0010107 | potassium ion import | 14.21 | 4 | 1.44E-04 | 3.80E-01 |
| GO:1990573 | potassium ion import across plasma membrane | 14.21 | 4 | 1.44E-04 | 3.17E-01 |
| GO:0055067 | monovalent inorganic cation homeostasis | 6.04 | 7 | 1.46E-04 | 2.74E-01 |
| GO:0015711 | organic anion transport | 3.73 | 11 | 1.66E-04 | 2.73E-01 |
| GO:0055075 | potassium ion homeostasis | 22.65 | 3 | 2.33E-04 | 3.42E-01 |
| GO:0071918 | urea transmembrane transport | 60.4 | 2 | 2.72E-04 | 3.58E-01 |
| GO:0006848 | pyruvate transport | 60.4 | 2 | 2.72E-04 | 3.26E-01 |
| GO:0015840 | urea transport | 60.4 | 2 | 2.72E-04 | 2.99E-01 |
| GO:1901475 | pyruvate transmembrane transport | 60.4 | 2 | 2.72E-04 | 2.76E-01 |
| GO:0045623 | negative regulation of T-helper cell differentiation | 20.13 | 3 | 3.46E-04 | 3.26E-01 |
| GO:0035728 | response to hepatocyte growth factor | 20.13 | 3 | 3.46E-04 | 3.04E-01 |
| GO:0015718 | monocarboxylic acid transport | 6.25 | 6 | 3.66E-04 | 3.01E-01 |
| GO:1902106 | negative regulation of leukocyte differentiation | 7.74 | 5 | 4.21E-04 | 3.26E-01 |
| GO:0034220 | ion transmembrane transport | 2.67 | 15 | 4.51E-04 | 3.30E-01 |
| GO:0072521 | purine-containing compound metabolic process | 3.53 | 10 | 5.14E-04 | 3.56E-01 |
| GO:0043371 | negative regulation of CD4-positive, alpha-beta T cell differentiation | 16.47 | 3 | 6.63E-04 | 4.37E-01 |
| GO:0031167 | rRNA methylation | 16.47 | 3 | 6.63E-04 | 4.16E-01 |
| GO:0009150 | purine ribonucleotide metabolic process | 3.67 | 9 | 7.42E-04 | 4.45E-01 |
| GO:0000466 | maturation of 5.8S rRNA from tricistronic rRNA transcript (SSU-rRNA, 5.8S rRNA, LSU-rRNA) | 40.27 | 2 | 8.07E-04 | 4.62E-01 |
| GO:0016338 | calcium-independent cell-cell adhesion via plasma membrane cell-adhesion molecules | 15.1 | 3 | 8.74E-04 | 4.80E-01 |
| GO:0002822 | regulation of adaptive immune response based on somatic recombination of immune receptors built from immunoglobulin superfamily domains | 5.25 | 6 | 9.34E-04 | 4.92E-01 |
| GO:0098742 | cell-cell adhesion via plasma-membrane adhesion molecules | 4.45 | 7 | 9.53E-04 | 4.83E-01 |

The table includes the GO term ID, description, the degree of enrichment, the number of differentially expressed genes associated with the GO term (*N*), as well as *P* and FDR-corrected *q*-values.

(*Leuciscus waleckii*) [13] indicated that there were few shared differentially expressed genes among all four ecotype pairs. Variation in gene expression was largely shaped by phylogeny rather than environment, and the shared differentially expressed genes among all four pairs showed strong variation in the direction and magnitude of differential expression across lineages. Overall, these results suggest that disparate lineages utilize different mechanisms for overcoming salinity challenges—at least at the level of gene expression. We found that various patterns of gene expression can emerge from crossing saltwater-freshwater boundaries, providing evidence of diverse responses of teleost lineages to a similar environmental challenge. A major question remaining is to what degree variation in gene expression in *L. perugiae* and the other lineages were shaped by phenotypic plasticity and by genetic differences in gene regulation. A previous study in stickleback found evidence for heritable gene expression differences between freshwater and saltwater ecotypes, evidence for shared plastic responses between ecotypes, but only little evidence for ecotype-specific plasticity [19]. Similar studies that combine field studies with laboratory experimentation are highly warranted for other study systems.

### Responses to variation in salinity in *L. perugiae*

**Regulation of transmembrane transport and gill epithelial permeability.** Overcoming the physiological challenges associated with transitioning between saltwater and freshwater environments requires modification of ion transport across the gill epithelia [16]. In our analysis of differentially expressed genes between a freshwater and hypersaline population of *L. perugiae*, we found evidence for differential regulation of ion transport. Among the genes that were upregulated in the hypersaline population, GO enrichment analysis revealed that several terms were associated with anion transport, sodium ion transport, metal ion transport, and maintaining chemical homeostasis. Genes corresponding to solute carrier families (e.g., *SLC9A3*, *SLC8B1*, *SLC12A8*, *SLC30A9*, *SLC7A1*) and ATPases (e.g., *ATP1B1*, *ATP6V1A*, *ATP13A3*) were among the upregulated genes that are important in maintaining ion and chemical homeostasis [59]. Increasing the rate of inorganic ion, amino acid, and nucleotide transport via upregulation of solute carriers allows aquatic organisms to maintain osmotic balance in saline environments, potentially facilitating acclimation or adaptation to hypersalinity [59]. Specifically, ATPases—such as $Na^+/K^+$-ATPase—have been well-studied for their role in salinity tolerance during both acclimation and adaptation [23, 31, 60–64]. Consistent with our study, previous Western blot analyses of freshwater and hypersaline *L. perugiae* populations have revealed an increase in $Na^+/K^+$-ATPase expression in hypersaline *L. perugiae* when compared to their freshwater conspecifics, which is essential for excreting $Na^+$ and $Cl^-$ out of the body to maintain homeostasis [23].

At very high salinity levels, it is especially difficult to pump $Na^+$ against the chemical gradient and out of the gill [65]. In a hypersaline environment, it therefore may be beneficial to maintain a lower epithelial salt permeability to avoid a back-flux of salts across the tight junctions of the gill epithelia [65, 66]. We found up-regulation of genes involved in cell-cell junction organization, including claudins involved in the formation of tight junctions (*CLDN3*, *CLDN4*, *CLDN5*, *CLDN8*, and *CLDN10*). Expression of gill claudin genes has been associated with salinity acclimation in fish [64, 66], and up-regulation of claudin-8 (*CLDN8*) has been shown to reduce the paracellular barrier permeability to $Na^+$ [67], suggesting a role for regulation of gill epithelial permeability via tight junctions during osmoregulatory challenges. Our findings support the hypothesis that modifications to transmembrane transport and gill epithelial permeability jointly contribute to the salt exclusion necessary for survival in a hypersaline environment.

**Cell cycle regulation and cellular metabolic and biosynthesis pathways.** Populations living in hypersaline environments must cope with the adverse effects of high salinity levels, often resulting in strategies for damage repair and energy reallocation [23, 62]. Functional analysis of the downregulated genes in the hypersaline population of *L. perugiae* showed signatures of downregulation of cell cycle and protein folding processes. Downregulation of genes involved in the cell cycle is expected to occur under high salinity stress to stop the replication of damaged cells and allow time for DNA repair [68, 69]. Similarly, locally adapted killifish living in freshwater and brackish environments exhibit evidence for divergence in the expression of genes underlying control of the cell cycle, supporting a central role in cell cycle regulation in adaptation to salinity stress [68, 70]. However, our findings may also suggest a general re-allocation of energy resources [68]. In hypersaline *L. perugiae*, the downregulation of genes involved in the cell cycle and protein folding could be reflective of an energetic trade-off between increased demand for processes involved in osmoregulation and the energetic cost of growth under stressful conditions [23]. In support of such energetic trade-offs, hypersaline *L. perugiae* have morphological differences from their freshwater counterparts, including a significantly smaller body size and reduced secondary sex features in males [23, 25].

Furthermore, energetic trade-offs associated with increased energy investment in osmoregulation in a hypersaline environment may also result in reduced investment in cellular metabolic and biosynthetic processes. In addition to genes involved in cell cycle processes, the weighted gene co-expression network analysis revealed that several cellular metabolic and biosynthetic processes were negatively associated with salinity (associated with genes in the blue module), including cellular aromatic compound metabolic process, organic cyclic compound metabolic process, nucleic acid metabolic process, cellular metabolic process, small molecule biosynthetic process, cellular macromolecule biosynthetic process, and carbohydrate biosynthetic process. Cellular metabolism and biosynthesis of compounds are energetically costly pathways for organisms [71], so these functions may receive less energy investment due to the high amounts of ATP required for ion transport and other osmoregulatory functions [68]. Future research will be important for identifying how osmoregulation in hypersaline environments may carry costs that influence cellular processes and how these costs impact organismal performance under different environmental conditions.

**Regulation of cell signaling.** To respond to osmoregulatory challenges, fishes must perceive their environmental salinity and maintain intracellular signals that modulate ion transport and other processes [72, 73]. We found differentially expressed genes associated with a variety of cell signaling processes. Several upregulated genes in the hypersaline population were enriched in GO terms associated with cell signaling, including signal transduction, regulation of signaling, G-protein-coupled receptor signaling pathway, and second-messenger-mediated signaling. G-protein-coupled receptor signaling is among the pathways involved in allowing aquatic organisms to sense their environmental salinity [74, 75]. Some of these signaling pathway components may also play a role in regulating ion transport, such as mitogen-activated protein kinases (MAPKs) and other serine/threonine protein kinases [76–79], which were among the upregulated genes. MAPK signaling pathways are also implicated in regulation of the cell cycle, some of which trigger cellular growth arrest and DNA damage repair in response to osmotic stress [79, 80]. The upregulation of signaling pathways, especially MAPK genes, may consequently play a role in the downregulation of cell-cycle genes that we found in the hypersaline *L. perugiae* population.

## Comparisons among replicated lineages

**Lineage-specific responses to variation in salinity.** Convergence in gene expression among independently evolved populations can occur in response to shared environmental stressors [81, 82]. Among fishes, cases of convergence in gene expression have been identified in response to selective pressures such as pollution [81], hydrogen sulfide [82], and the absence of light in caves [83]. In contrast, our comparative transcriptomics analysis indicated there is little evidence for convergence in gene expression patterns among fish lineages in response to variation in salinity. Even among the shared differentially expressed genes, there is considerable variation in the magnitude and direction of expression differences between ecotypes across lineages. This finding mirrors genomic analyses that looked for convergent signatures associated with adaptation to salinity variation, which found osmoregulation genes to be common targets of selection but also variation in the exact genes that were involved across different lineages [84].

Our finding of lineage-specific responses to variation in salinity, and therefore low levels of convergence in gene expression, could be explained by several non-mutually exclusive hypotheses [85]. First, changes in both protein-coding DNA sequences and gene expression may occur during adaptation to a novel environment, or they may occur independently [55, 86, 87]. Under certain conditions, selection may favor changes in protein structure and function

via modifications at the sequence level, regardless of whether or not gene expression is affected [86]. Alternatively, changes in gene expression may be favored without selection acting at the sequence level [86]. Variation in how selection acts on gene expression could contribute to the lack of convergence among the lineages in our study, and there also may be stronger signals of convergence at levels other than gene expression.

Secondly, differences in genetic architecture among lineages may cause different responses to selection, leading to diverse evolutionary outcomes [85]. Convergence is therefore less likely to occur with increasing genetic divergence between lineages [88], so it is not necessarily surprising that we did not find convergence among distantly related fishes. Even if the genetic architectures are similar among lineages, idiosyncratic responses to selection can also arise as a consequence of functional redundancy [89]. Specifically, modification of different genes and pathways may have equivalent functional and fitness consequences [78, 84]. Functional redundancy is particularly common in complex traits like salinity tolerance that involve many genes and physiological pathways [89]. While salinity tolerance is a shared outcome among the lineages in our analysis, the molecular mechanisms underlying osmoregulation may be unique to each lineage.

Third, the directionality of habitat transition may impact gene expression responses, especially if there are genetic adaptations to the habitat of origin. For example, a high-activity version of an ion transporter may be downregulated during transitions to freshwater, while a low-activity version of the same enzyme may be upregulated in the opposite direction. Among the four lineages we included in our study, two of them transitioned from saltwater to freshwater environments (*Gasterosteus* and *Odontesthes*), and two transitioned from freshwater to saltwater environments (*Limia* and *Leuciscus*). If the direction of transition elicits similar responses, then we would expect populations that transitioned in the same direction to share more differentially expressed genes than those that did not transition in the same direction. However, we did not find this to be the case, as the pairs transitioning in the same direction share fewer differentially expressed genes than those that made opposite transitions (Fig 3C).

Finally, covariation with other sources of selection—both abiotic and biotic—may cause idiosyncratic gene expression patterns across lineages. Beyond the challenges directly imposed by variation in salinity, such habitat transitions are often accompanied by other environmental challenges, such as variation in temperature, exposure to novel parasites, and restructuring of host-associated microbial communities [2, 20, 90, 91]. For example, hypersaline Amur ide must cope with high alkalinity stress in addition to salinity, and hypersaline *L. perugiae* have a warmer environment than freshwater *L. perugiae* [13, 23]. Additionally, there are differences in the gill microbial communities of saltwater and freshwater populations of South American silversides [20]. Such correlated environmental factors can contribute variation in gene expression responses to salinity transitions we observed.

**Shared responses involving ion transport and immune function.** Although there is little evidence for convergence among all four lineage pairs included in our analysis, there was some functional overlap in the differentially expressed genes shared among three or more lineages. Most of the shared differentially expressed genes were associated with ion transport and immune system processes. Genes implicated in ion transport and immune system processes were also identified among the ten differentially expressed genes that were shared among all four lineages. For example, *SLC9A3*, a solute carrier gene involved in osmoregulation, was among the ten shared genes, suggesting its role in salinity transitions among fishes [59]. Another shared differentially expressed gene, *CEACAM1* (Carcinoembryonic antigen-related cell adhesion molecule 1), has been implicated in immune processes in other systems and is thought to be involved in a variety of pathways, but its function is not well known [92, 93]. As previously discussed, regulation of ion transport is expected to be crucial when crossing a

saltwater-freshwater boundary [16], and osmoregulation genes also exhibit evidence of convergent evolution during salinity transitions [84].Variation in expression of immune genes, particularly those related to inflammation and adaptive immunity, has also been documented in fish during salinity acclimation [94, 95]. Under a variety of selection pressures, locally adapted populations also frequently show divergence in immune genes due to other factors such as life history differences, different parasite exposure, and shifts in the microbiome [20, 95–98]. Immune loci are consequently evolutionary hotspots in diversification [99–101]. It needs to be tested whether changes in the expression of immune-related genes are directly linked to variation in salinity or whether these genes generally respond to changes in correlated biotic sources of selection.

Overall, our analysis of gene expression patterns between locally adapted freshwater and hypersaline populations of *L. perugiae* provide insight into how this livebearing fish maintains homeostasis in a hypersaline environment. In addition, comparisons between four population pairs of freshwater and saltwater ecotypes in disparate teleost lineages showed little evidence for convergence, as there were only ten differentially expressed genes that were shared among them all. Despite this, we found that the differentially expressed genes shared in three or more of the lineages reflected biological processes related to ion transport and immune functioning. Our findings provide insight into shared and unique gene expression responses to salinity variation, broadly informing our understanding of salinity tolerance in aquatic organisms. Furthermore, comparisons in gene expression across species that have made habitat transitions are important for understanding mechanisms of adaptation to novel environments. Our results shed light on the repeatability of transcriptomic responses to salinity variation among fishes, which remains a relatively underexplored area of research despite its relevance for aquatic ecology and evolutionary biology. Future research comparing gene expression in fishes from freshwater and saltwater environments in both laboratory and field settings will be important for identifying the degree in which shared or unique gene expression responses are due to plasticity or adaptation, and additional studies that take a comparative transcriptomic approach with more lineages—representing transitions from freshwater to saltwater and vice versa as well as spanning a range of transition time and divergence time between populations —will provide further insight into shared mechanisms of osmoregulation among fishes.

## Supporting information

**S1 Fig. Biological processes from the GO enrichment analysis that were associated with upregulated genes in the saltwater population of *L. perugiae*.**
(PNG)

**S2 Fig. Biological processes from the GO enrichment analysis that were associated with downregulated genes in the saltwater population of *L. perugiae*.**
(PNG)

**S3 Fig. Biological processes from the GO enrichment analysis that were associated with differentially expressed genes shared among three or more lineages.**
(PNG)

**S1 Appendix. Contains Tables A-E, which include the mapping statistics for all four population pairs, the weighted gene co-expression network analysis (WGCNA) results, and the significantly enriched GO terms from the WGCNA modules.**
(XLSX)

## Acknowledgments

We thank Ingo Schlupp for help with the field collections and the constructive feedback on the manuscript. We thank the authors of past salinity transcriptomics papers [13, 19, 20] who made their data publicly available and enabled the comparative analyses presented here.

## Author Contributions

**Conceptualization:** Elizabeth J. Wilson, Nick Barts, John L. Coffin, James B. Johnson, Carlos M. Rodríguez Peña, Joanna L. Kelley, Michael Tobler, Ryan Greenway.

**Data curation:** Joanna L. Kelley, Ryan Greenway.

**Formal analysis:** Elizabeth J. Wilson, Nick Barts, John L. Coffin, Joanna L. Kelley, Michael Tobler, Ryan Greenway.

**Funding acquisition:** Joanna L. Kelley, Michael Tobler.

**Investigation:** James B. Johnson, Carlos M. Rodríguez Peña, Michael Tobler.

**Project administration:** Michael Tobler.

**Resources:** Carlos M. Rodríguez Peña.

**Supervision:** Michael Tobler, Ryan Greenway.

**Visualization:** Elizabeth J. Wilson, Michael Tobler.

**Writing – original draft:** Elizabeth J. Wilson, Michael Tobler.

**Writing – review & editing:** Elizabeth J. Wilson, Nick Barts, John L. Coffin, James B. Johnson, Carlos M. Rodríguez Peña, Joanna L. Kelley, Michael Tobler, Ryan Greenway.

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
