## [Decision Letter · Decision Letter 0]

12 Feb 2024

PONE-D-23-43144Gene expression signatures of salinity transitions in Limia perugiae (Poeciliidae), with comparisons to other teleostsPLOS ONE

Dear Dr. Tobler,

Thank you for submitting your manuscript to PLOS ONE. After careful consideration, we feel that it has merit but does not fully meet PLOS ONE’s publication criteria as it currently stands. Therefore, we invite you to submit a revised version of the manuscript that addresses the points raised during the review process.

1. The data cannot support the conclusions. PLOS ONE is designed to communicate primary scientific research, and welcome submissions in any applied discipline that will contribute to the base of scientific knowledge. But the data of this manuscript cannot support the conclusions.

2. This manuscript needs to adhere the PLOS Data Policy. The authors need to make all methods, materials and data underlying the findings in their manuscript fully available.

3. The revised manuscript needs to address each of the comments of the reviewers.

We look forward to receiving your revised manuscript.

Kind regards,

Tzong-Yueh Chen, Ph.D.

Academic Editor

PLOS ONE

Journal Requirements:

3. To comply with PLOS ONE submissions requirements, in your Methods section, please provide additional information regarding the experiments involving animals and ensure you have included details on (1) methods of sacrifice, (2) methods of anesthesia and/or analgesia, and (3) efforts to alleviate suffering.

"Funding was provided by National Science Foundation ( IOS-1931657, IOS-2311366), the Army Research Office (W911NF-15-1-0175, W911NF-16-1-0225), and the Des Lee Collaborative Vision in Zoological Studies. "

"We thank Ingo Schlupp for help with the field collections and the constructive feedback on the manuscript. Funding was provided by National Science Foundation ( IOS-1931657, IOS-2311366), the Army Research Office (W911NF-15-1-0175, W911NF-16-1-0225), and the Des Lee Collaborative Vision in Zoological Studies. We thank the authors of past salinity transcriptomics papers (Hughes et al. 2017; Gibbons et al. 2017; Xu et al. 2013) who made their data publicly available and enabled the comparative analyses presented here."

"Funding was provided by National Science Foundation ( IOS-1931657, IOS-2311366), the Army Research Office (W911NF-15-1-0175, W911NF-16-1-0225), and the Des Lee Collaborative Vision in Zoological Studies. "

7. Please provide a complete Data Availability Statement in the submission form, ensuring you include all necessary access information or a reason for why you are unable to make your data freely accessible. If your research concerns only data provided within your submission, please write "All data are in the manuscript and/or supporting information files" as your Data Availability Statement.

8. Please include your full ethics statement in the ‘Methods’ section of your manuscript file. In your statement, please include the full name of the IRB or ethics committee who approved or waived your study, as well as whether or not you obtained informed written or verbal consent. If consent was waived for your study, please include this information in your statement as well. 

Reviewers' comments:

Reviewer's Responses to Questions

**Comments to the Author**

1. Is the manuscript technically sound, and do the data support the conclusions?

Reviewer #1: Yes

Reviewer #2: Yes

Reviewer #3: Yes

2. Has the statistical analysis been performed appropriately and rigorously? 

Reviewer #1: Yes

Reviewer #2: Yes

Reviewer #3: Yes

3. Have the authors made all data underlying the findings in their manuscript fully available?

Reviewer #1: Yes

Reviewer #2: No

Reviewer #3: Yes

4. Is the manuscript presented in an intelligible fashion and written in standard English?

Reviewer #1: Yes

Reviewer #2: Yes

Reviewer #3: Yes

5. Review Comments to the Author

Reviewer #1: I have thoroughly reviewed the manuscript concerning the gene expression changes in Limia perugiae during transitions between freshwater and saltwater environments. This study employs RNA sequencing to elucidate the genomic responses associated with environmental salinity changes, which is a critical aspect of aquatic biology. Researchers are carefully looking at how gene expression is different in hypersaline and freshwater populations of Limia perugiae. They want to find out how this species can tolerate high salt levels. The manuscript is well-structured, detailing the experimental approach, data analysis, and interpretation of results, reflecting the scientific rigor and depth of the research.

The study's findings, focusing on genes involved in ion transport, chemical homeostasis, and cell signaling, are intriguing and contribute significantly to our understanding of salinity adaptation in fish. Furthermore, the comparative analysis with other teleosts provides valuable insights into the evolutionary aspects of salinity tolerance. These contributions are noteworthy and enrich the field of ichthyology. In the following, several significant points should be revised in this manuscript:

Introduction:

(1) The experimental materials were not adequately introduced in the background. The selection of pearl trevally as a model organism and a concise explanation of its biological attributes should be provided.

(2) The article's structure may use some clarification. To facilitate readers' comprehension of the article's structure and primary arguments, straightforward subheadings such as "research gaps" and "objectives" may be implemented.

(3) Certain assertions may be articulated with greater agility and precision. For example, the first sentence, “Variation in salinity imposes osmoregulatory challenges on aquatic organisms,” could be changed to “Salinity variation imposes osmoregulatory challenges for aquatic organisms.”.

(4) The study's objective and novelty may be expounded upon. However, the study's specific purpose, content, and innovation are insufficiently clear.

Materials and Methods:

(1) The study lacks measurements of background environmental parameters, including salinity levels and physico-chemical metrics, such as pH, at the sites where experimental animals were collected. Quantifying these parameters would provide important context about the habitat conditions.

(2) The description of the RNA extraction and library construction methodology is fairly concise. Elaborating on the experimental procedures and specifics would enhance reproducibility.

(3) Additional details can be provided when outlining the statistical workflow for the differential gene expression analysis, particularly regarding dispersion estimation approaches, design matrix building, etc. Expanding on these analytical steps may improve transparency.

(4) The rationale behind parameter selection for the co-expression network analysis should be explained further, notably the criteria for choosing the soft thresholding power. Describing the decision process more thoroughly would augment the methodology.

(5) In the comparative analysis between species, conveying quality control measures such as homology assessments would strengthen the integrity of the cross-species alignments. Justifying the selection of certain analytical parameters would also enhance robustness.

Results:

(1) The functional enrichment analysis results could benefit from more visual representations, such as bar charts illustrating significantly enriched GO terms, rather than just tables. While informative, tabular-only reporting reduces readability. Utilizing straightforward data visualization techniques can aid in conveying complex results.

(2) The visualization of the 10 shared differential genes from the cross-species comparative analysis (Fig. 4) provides limited information. Incorporating quantitative outcomes for each species, such as comparisons of differential expression directionality and magnitude, would better showcase this data.

(3) The reporting format for the functional enrichment outcomes in the multi-species comparative section is inconsistent with other segments, lacking statistical details like P-values and FDR corrections. Formatting should be unified for clarity.

(4) Certain results around candidate environmental response pathways are repeated in the Discussion. Reducing the reiteration of key data in the Discussion could improve conciseness.

Discussion:

(1) The discussion should directly address how the major findings relate to the original hypotheses and research questions stated in the introduction. Explicitly connecting the conclusions to the study goals would strengthen the framing.

(2) A number of the unique GO terms enriched in the blue module pertain to cellular metabolism and biosynthetic pathways. However, discussions integrating energy metabolism analyses and adaptive capacity were notably absent, despite their relevance. Many transmembrane ion channels critically rely on ATP to function, representing substantial energetic costs. Considering the many complex processes that depend on ion and transporter activity, it would have been helpful to look at metabolic needs and trade-offs along with the processes themselves. This absence of in-depth discussion into the interdependency of bioenergetics and osmoregulation, unfortunately, represents a missed opportunity within the study. Direct analyses quantifying such relationships should be prioritized in this work.

(3) The observed instances of consistent differential expression across a few genes warrant further interpretation. The discussion would benefit from speculating upon why genes integral to these key processes did not demonstrate expected co-occurrence patterns.

(4) When comparing multi-species differences in differential expression, the dialogue on potential explanatory factors like life history and genetic structure distinctions feels disconnected. Weaving this into the overarching narrative flow could provide clearer context.

(5) Considering the genetic relationships between the study populations is advised when evaluating whether sympatric groups exhibit substantial genetic differentiation. Incorporating this aspect lends broader relevance.

(6) Commenting on whether environmental cues elicited plastic or genetically-programmed gene expression changes would add an epigenetic dimension currently lacking.

(7) Elaborating on the implications of the discoveries for the fields of evolutionary biology and aquatic ecology would highlight the potential wider impacts. Discussing how the results advance the current understanding of gene expression in salinity transitions would also be insightful.

(8) While limitations are noted, more fully detailing study constraints and avenues for future research, like repeatability and comparative pursuits, would enhance the scholarship and constructive criticism.

Reviewer #2: I have completed a thorough review of the manuscript titled "Gene expression signatures of salinity transitions in Limia perugiae (Poeciliidae), with comparisons to other teleosts" by Elizabeth J. Wilson et al., submitted for consideration in PLoS One (Manuscript ID: PONE-D-23-43144). Overall, the study provides valuable insights into the gene expression patterns associated with salinity transitions in Limia perugiae. However, I have identified several points that require clarification or further attention:

Title Clarification:

Given the study involves Limia perugiae populations from a hypersaline lagoon and a freshwater stream, rather than transitions between seawater and freshwater, the title may benefit from clarification. I recommend considering a title such as "Gene expression signatures in Limia perugiae populations from hypersaline and freshwater habitats" to accurately reflect the environmental context.

Salinity Percentage in Hypersaline Lagoon:

The authors are encouraged to provide information on the salinity percentage in the hypersaline lagoon (Laguna Oviedo: 17.801° N, 71.363° W). Clarifying the salinity range or percentage in the hypersaline lagoon is crucial for a comprehensive understanding of the study system.

SRA Accessions in Table 1:

In Table 1, the term "Pending" under the SRA Accessions column needs to be addressed. I recommend that the authors provide the specific SRA Accessions or update the information to ensure the reproducibility and accessibility of their RNA-seq data.

Explanation of Abbreviations in Figure 2(A):

In Figure 2(A), clarification is needed for the abbreviations "DC" and "MD" associated with the color bars below the plot. The authors should explicitly state the meanings of these abbreviations, aiding readers in the accurate interpretation of the data presented.

Confirmation of Transcriptomic Data through Real-time qPCR:

The manuscript mentions that confirmation of the transcriptomic data through real-time quantitative PCR (qPCR) is pending. I suggest the authors emphasize the importance of validating their RNA-seq results using qPCR and discuss how this confirmation would strengthen the reliability of their findings.

In addition to these specific points, I commend the authors for their comprehensive analysis of gene expression patterns in Limia perugiae and the comparison with other teleosts. The study contributes significantly to our understanding of salinity transitions in fish. Addressing the mentioned concerns will further enhance the clarity, reproducibility, and overall quality of the manuscript.

Reviewer #3: In this study, the author and the research team devoted extensive effort to the collection of samples across a broad scope. I greatly admire the dedication demonstrated by the author in undertaking this research.

Following the collection of samples from various regions, the authors conducted transcriptome sequencing and extensive comparative analysis, a task of considerable difficulty. The scope of sampling was broad and intricate, and transcriptome sequencing required substantial resources as well as meticulous statistical analysis. This highlights the authors' dedication and contribution to this research.

Furthermore, the research team dedicated significant time to comparing transcriptomic expressions among different fish species and conducted extensive statistical analyses in an attempt to identify whether similar mechanisms exist across different phylogenies in the regulation of salinity. However, the conclusions drawn suggest that teleosts from diverse phylogenies may employ distinct regulatory mechanisms for salinity acclimation. I highly commend the statistical and analytical efforts of the research team. This study provides a valuable direction for future researchers investigating the genetic regulation of euryhaline fish in response to salinity changes.

However, I posit that the author and research team should consider several critical aspects in future studies aimed at examining the regulatory mechanisms and acclimatization of euryhaline fish species across different salinity environments. Firstly, the native habitat of the species, whether seawater or freshwater, is vital, as fish originating from seawater acclimating to freshwater conditions and those from freshwater acclimating to seawater conditions may utilize distinct regulatory mechanisms. Secondly, if the species has been residing in either freshwater or seawater for an extended period, their regulatory mechanisms may have already adapted to those specific environments, rather than being amenable to acclimatization across varying salinity levels.

I express my gratitude towards the authors and the research team for their efforts and contributions to this manuscript and recommend the publication of this original research article.

6. PLOS authors have the option to publish the peer review history of their article (what does this mean?). If published, this will include your full peer review and any attached files.

Reviewer #1: **Yes: **Yung-Che Tseng

Reviewer #2: No

Reviewer #3: No

---

## [Author Response · Author response to Decision Letter 0]

2 Sep 2024

We have uploaded responses to reviewers as a separate file. Below is the copy/pasted version of that text:

We thank the editor and reviewers for their thorough evaluation of our research and for their comments to improve the manuscript. Under each comment from the editor and reviewers that required addressing, we have detailed our edits to the manuscript and our responses, which are provided below.

Journal Requirements:

We have checked all styles requirements in the process of the revision. 

We included a statement with permit information (lines 143-145).

3. To comply with PLOS ONE submissions requirements, in your Methods section, please provide additional information regarding the experiments involving animals and ensure you have included details on (1) methods of sacrifice, (2) methods of anesthesia and/or analgesia, and (3) efforts to alleviate suffering.

We have included the method of sacrifice (lines 141-142) and information about the approval for all procedures involving animals from the Institutional Animal Care and Use Committee of Kansas State University (lines 145-146).

We will made sure that all grant numbers match.

"Funding was provided by National Science Foundation ( IOS-1931657, IOS-2311366), the Army Research Office (W911NF-15-1-0175, W911NF-16-1-0225), and the Des Lee Collaborative Vision in Zoological Studies. "

We added a corresponding statement to the text.

"We thank Ingo Schlupp for help with the field collections and the constructive feedback on the manuscript. Funding was provided by National Science Foundation ( IOS-1931657, IOS-2311366), the Army Research Office (W911NF-15-1-0175, W911NF-16-1-0225), and the Des Lee Collaborative Vision in Zoological Studies. We thank the authors of past salinity transcriptomics papers (Hughes et al. 2017; Gibbons et al. 2017; Xu et al. 2013) who made their data publicly available and enabled the comparative analyses presented here."

"Funding was provided by National Science Foundation ( IOS-1931657, IOS-2311366), the Army Research Office (W911NF-15-1-0175, W911NF-16-1-0225), and the Des Lee Collaborative Vision in Zoological Studies. "

We have removed all funding-related text from the manuscript.

7. Please provide a complete Data Availability Statement in the submission form, ensuring you include all necessary access information or a reason for why you are unable to make your data freely accessible. If your research concerns only data provided within your submission, please write "All data are in the manuscript and/or supporting information files" as your Data Availability Statement.

We will provide a complete Data Availability Statement in the submission form.

8. Please include your full ethics statement in the ‘Methods’ section of your manuscript file. In your statement, please include the full name of the IRB or ethics committee who approved or waived your study, as well as whether or not you obtained informed written or verbal consent. If consent was waived for your study, please include this information in your statement as well. 

We added an ethics statement to the methods (lines 143-146).

We added captions for our Supporting Information files in a “Supporting information” section at the end of the manuscript (lines 921-930), and we updated the supporting information file names within the text of the manuscript.

Reviewer Requirements:

Reviewer #1: I have thoroughly reviewed the manuscript concerning the gene expression changes in Limia perugiae during transitions between freshwater and saltwater environments. This study employs RNA sequencing to elucidate the genomic responses associated with environmental salinity changes, which is a critical aspect of aquatic biology. Researchers are carefully looking at how gene expression is different in hypersaline and freshwater populations of Limia perugiae. They want to find out how this species can tolerate high salt levels. The manuscript is well-structured, detailing the experimental approach, data analysis, and interpretation of results, reflecting the scientific rigor and depth of the research.

The study's findings, focusing on genes involved in ion transport, chemical homeostasis, and cell signaling, are intriguing and contribute significantly to our understanding of salinity adaptation in fish. Furthermore, the comparative analysis with other teleosts provides valuable insights into the evolutionary aspects of salinity tolerance. These contributions are noteworthy and enrich the field of ichthyology. In the following, several significant points should be revised in this manuscript:

Introduction:

(1) The experimental materials were not adequately introduced in the background. The selection of pearl trevally as a model organism and a concise explanation of its biological attributes should be provided.

In this study, we did not study pearl trevally. We focused on Limia perugiae to better understand salinity tolerance in fish of this family (Poeciliidae), since Limia perugiae occurs naturally in both freshwater and hypersaline environments, but L. perugiae otherwise does not have attributes of a model organism outside of our proposed use of this species to investigate salinity tolerance. We also added a subheading titled “Salinity tolerance in Limia perugiae (Poeciliidae)” (line 104) to improve clarity on studying Limia perugiae in the context of understanding salinity tolerance in fishes.

(2) The article's structure may use some clarification. To facilitate readers' comprehension of the article's structure and primary arguments, straightforward subheadings such as "research gaps" and "objectives" may be implemented.

Throughout the introduction, we added subheadings (titled “Salinity transitions in fishes” (line 61) “Gene expression responses to variation” (line 87), “Salinity tolerance in Limia perugiae (Poeciliidae)” (line 104), and “Objectives” (line 123)) to improve clarity of the manuscript’s structure. 

(3) Certain assertions may be articulated with greater agility and precision. For example, the first sentence, “Variation in salinity imposes osmoregulatory challenges on aquatic organisms,” could be changed to “Salinity variation imposes osmoregulatory challenges for aquatic organisms.”.

We re-worded the first sentence to be changed to “Salinity variation imposes osmoregulatory challenges on aquatic organisms” (line 51).

(4) The study's objective and novelty may be expounded upon. However, the study's specific purpose, content, and innovation are insufficiently clear.

We included the importance and novelty of investigating patterns of convergence in gene expression among fish lineages that have made salinity transitions in the newly titled section “Gene expression responses to salinity variation” (lines 87-102). We also included the importance and novelty of studying gene expression involved in salinity tolerance of Limia perugiae (lines 104-121). In the section now titled “Objectives,” we include the specific approach and objectives of our study (lines 123-134).

Materials and Methods:

(1) The study lacks measurements of background environmental parameters, including salinity levels and physico-chemical metrics, such as pH, at the sites where experimental animals were collected. Quantifying these parameters would provide important context about the habitat conditions.

We do not have pH measurements available, but we did add the salinity levels from the hypersaline and freshwater habitats where we collected L. perugiae to the methods section to provide more information about these environments (lines 140-141).

(2) The description of the RNA extraction and library construction methodology is fairly concise. Elaborating on the experimental procedures and specifics would enhance reproducibility.

We added information to this section detailing the specific procedures that we implemented for library preparation (lines 150-153).

(3) Additional details can be provided when outlining the statistical workflow for the differential gene expression analysis, particularly regarding dispersion estimation approaches, design matrix building, etc. Expanding on these analytical steps may improve transparency.

We have added additional information on the statistical approach (lines 186-190).

(4) The rationale behind parameter selection for the co-expression network analysis should be explained further, notably the criteria for choosing the soft thresholding power. Describing the decision process more thoroughly would augment the methodology.

We added information referencing the detailed protocols we used to select parameters for the co-expression network analysis. Specifically, to pick a soft thresholding power value that ensured our network fit the approximate scale free topology criterion while retaining the highest possible mean connectivity between the network genes, we followed the recommendations published by Zhang and Horvath (2005) that explain how to use the pickSoftThreshold function in the WGCNA package to pick the smallest soft thresholding power that fits both criteria. This section now includes the reference to the suggestions we followed in Zhang and Horvath (2005) for parameter selection, and we also clarified that this soft thresholding power is the smallest possible value that satisfies the required criteria for the co-expression network models (lines 214-219).

(5) In the comparative analysis between species, conveying quality control measures such as homology assessments would strengthen the integrity of the cross-species alignments. Justifying the selection of certain analytical parameters would also enhance robustness.

To make comparisons across species, we used OrthoFinder v2.2.6 to identify orthologous genes among the reference genomes, which is a standard tool for determining orthologous genes. We added additional details regarding the parameters that were used in running OrthoFinder and the number of orthologous groups of genes (lines 265-273). 

Results:

(1) The functional enrichment analysis results could benefit from more visual representations, such as bar charts illustrating significantly enriched GO terms, rather than just tables. While informative, tabular-only reporting reduces readability. Utilizing straightforward data visualization techniques can aid in conveying complex results.

We added visual representations for all of the functional enrichment analysis results. Specifically, we added two supplementary figures that show the GO processes for the upregulated and downregulated genes in the hypersaline environment in the Limia perugiae portion of the manuscript (S1 and S2 Figs), as well as a supplementary figure showing the GO processes for the differentially expressed genes shared in three or more lineages in the comparative portion of the manuscript (S3 Fig).

(2) The visualization of the 10 shared differential genes from the cross-species comparative analysis (Fig. 4) provides limited information. Incorporating quantitative outcomes for each species, such as comparisons of differential expression directionality and magnitude, would better showcase this data.

We chose to visualize the expression variation among the shared differentially expressed genes of all the lineages, rather than show the differentially expressed genes for each species, because these results were previously reported in the studies from which we pulled the data for the cross-species analyses in this paper. Since quantitative outcomes for the species included in our manuscript have been reported in previous studies, we chose to not repeat those outcomes here and instead focused on comparative analysis of all the lineages.

(3) The reporting format for the functional enrichment outcomes in the multi-species comparative section is inconsistent with other segments, lacking statistical details like P-values and FDR corrections. Formatting should be unified for clarity.

The tables for the functional enrichment results for the L. perugiae segment and the comparative section include the same statistical details and column headings (Tables 2 and 4), and we clarified the P-values and FDR-corrected P-values in the text so that the multi-species comparative section is consistent with the text of the L. perugiae section (lines 408-409 and line 413). We also updated the title for Table 4 to be consistent with the text in the results that mentions the P-values and FDR corrections (lines 414-416). 

(4) Certain results around candidate environmental response pathways are repeated in the Discussion. Reducing the reiteration of key data in the Discussion could improve conciseness.

Throughout the discussion, we reduced the number of instances where candidate environmental response pathways are repeated, which allowed us to expand on other topics in the discussion that were suggested in the reviewer comments below.

Discussion:

(1) The discussion should directly address how the major findings relate to the original hypotheses and research questions stated in the introduction. Explicitly connecting the conclusions to the study goals would strengthen the framing.

We re-worded the beginning of the discussion to relate it back to the original aims and research questions stated in the introduction (lines 422-425). 

(2) A number of the unique GO terms enriched in the blue module pertain to cellular metabolism and biosynthetic pathways. However, discussions integrating energy metabolism analyses and adaptive capacity were notably absent, despite their relevance. Many transmembrane ion channels critically rely on ATP to function, representing substantial energetic costs. Considering the many complex processes that depend on ion and transporter activity, it would have been helpful to look at metabolic needs and trade-offs along with the processes themselves. This 

---

## [Decision Letter · Decision Letter 1]

20 Nov 2024

Gene expression signatures betweenLimia perugiae (Poeciliidae) populations from freshwater and hypersaline habitats, with comparisons to other teleosts

PONE-D-23-43144R1

Dear Dr. Tobler,

We’re pleased to inform you that your manuscript has been judged scientifically suitable for publication and will be formally accepted for publication once it meets all outstanding technical requirements.

Kind regards,

Tzong-Yueh Chen, Ph.D.

Academic Editor

PLOS ONE

Additional Editor Comments (optional):

Reviewers' comments:

Reviewer's Responses to Questions

**Comments to the Author**

1. If the authors have adequately addressed your comments raised in a previous round of review and you feel that this manuscript is now acceptable for publication, you may indicate that here to bypass the “Comments to the Author” section, enter your conflict of interest statement in the “Confidential to Editor” section, and submit your "Accept" recommendation.

Reviewer #2: All comments have been addressed

2. Is the manuscript technically sound, and do the data support the conclusions?

Reviewer #2: Yes

3. Has the statistical analysis been performed appropriately and rigorously? 

Reviewer #2: Yes

4. Have the authors made all data underlying the findings in their manuscript fully available?

Reviewer #2: Yes

5. Is the manuscript presented in an intelligible fashion and written in standard English?

Reviewer #2: Yes

6. Review Comments to the Author

Reviewer #2: Based on the revisions and responses provided, I find the manuscript acceptable for publication. The study contributes valuable knowledge to the field of salinity tolerance in teleosts and provides a well-rounded approach to gene expression analysis across environmental gradients. Thank you for your thorough work and for addressing the previous concerns so effectively.

7. PLOS authors have the option to publish the peer review history of their article (what does this mean?). If published, this will include your full peer review and any attached files.

Reviewer #2: No

---

## [Editor Report · Acceptance letter]

26 Nov 2024

PONE-D-23-43144R1 

PLOS ONE

Dear Dr. Tobler, 

I'm pleased to inform you that your manuscript has been deemed suitable for publication in PLOS ONE. Congratulations! Your manuscript is now being handed over to our production team.

Kind regards, 

on behalf of

Prof. Tzong-Yueh Chen 

Academic Editor

PLOS ONE